# Anti-Inflammatory Properties of Plants from Serbian Traditional Medicine

**DOI:** 10.3390/life13040874

**Published:** 2023-03-24

**Authors:** Katarina Radovanović, Neda Gavarić, Milica Aćimović

**Affiliations:** 1Department of Pharmacy, Faculty of Medicine, University of Novi Sad, Hajduk Veljkova 3, 21000 Novi Sad, Serbia; 2Department of Vegetable and Alternative Crops, Institute of Field and Vegetable Crops Novi Sad, Maksima Gorkog 30, 21000 Novi Sad, Serbia

**Keywords:** traditional medicine, anti-inflammatory properties, ethnopharmacology

## Abstract

Inflammation is a natural protective response of the human body to a variety of hostile agents and noxious stimuli. Standard anti-inflammatory therapy includes drugs whose usage is associated with a number of side effects. Since ancient times, natural compounds have been used for the treatment of inflammation. Traditionally, the use of medicinal plants is considered safe, inexpensive, and widely acceptable. In Serbia, traditional medicine, based on the strong belief in the power of medicinal herbs, is the widespread form of treatment. This is supported by the fact that Serbia is classified as one of 158 world centers of biodiversity, which confirms that this country is a treasure of medicinal herbs. Some of the most used herbs for the treatment of inflammations of various causes in Serbian tradition are yarrow, common agrimony, couch grass, onion, garlic, marshmallow, common birch, calendula, liquorice, walnut, St. John’s wort, chamomile, peppermint, white willow, sage, and many others. The biological activity and anti-inflammatory effect of selected plants are attributed to different groups of secondary biomolecules such as flavonoids, phenolic acids, sterols, terpenoids, sesquiterpenes, and tannins. This paper provides an overview of plants with traditional anti-inflammatory use in Serbia with reference to available studies that examined this effect. Plants used in traditional medicine could be a powerful source for the development of new remedies. Therefore intensive research on the bioactive potential of medicinal plants in each region should be the focus of scientists around the world.

## 1. Introduction

Inflammation is the body’s automatic protective defence reaction to tissue injury or the invasion of foreign factors (toxins and pathogens) [1]. A controlled inflammatory response is an important beneficial process that is part of the maintained normal homeostasis of tissue [2]. The duration and extent of the inflammatory response are of key importance for its outcome and consequences. The process of acute inflammation includes phagocytosis, apoptosis, or activations of pro-inflammatory mediators that leads to the clearance of injurious stimuli and restore normal physiology [3]. However, chronic inflammation is not a useful process and it causes various pathological disorders including Alzheimer’s disease, cancer, rheumatoid arthritis, type 2 diabetes, and obesity, as well as a cardiovascular and pulmonary disease [4]. Chronic disorders are regarded as a leading cause of death globally, with 60% of these deaths due to chronic inflammatory diseases [5]. These inflammatory changes are induced by cytokines and other inflammatory mediators. Cytokines are classified into two major categories: pro-inflammatory and anti-inflammatory cytokines. Several cytokines including interleukin (IL)-1, tumour necrosis factor-alpha (TNF-a), IL-6 and IL-8, and chemokines such as granulocyte colony-stimulating factor (G-CSF) and granulocyte-macro-phage colony-stimulating factor (GM-CSF) play a key role in acute inflammatory reactions [6].

A conventional therapeutic option for the treatment of inflammation and associated pain is nonsteroidal anti-inflammatory drugs, but their use is associated with a multitude of unwanted effects. For this reason, new research is focused on the search for safe natural substitutes for conventional anti-inflammatory drugs. The latest research has determined numerous pharmacological targets including cytokines, chemokines, transcription factors, complement activation pathways, eicosanoids, reactive oxygen species (ROS), and reactive nitrogen species (RNS) [4,7].

Since ancient times, people have relied on medicinal plants in the treatment of various health disorders, including inflammation and its complications. These actions are attributed to the complex chemical composition and the presence of secondary biomolecules in plants such as phenolic compounds, flavonoids, saponins, and sesquiterpenes [8]. The challenge of science is to find the exact chemical compound responsible for the observed pharmacological effects [9]. Moreover, it is important to recognize the side effects and potential interactions between medicinal plants and other synthetic drugs [10].

Bearing in mind that for centuries all nations have developed their traditional medicine based on the plants that grow in their environment and that precisely traditional methods of treatment have become very popular in recent years, and ubiquitous in everyday life, the aim of this paper was to review the plants with anti-inflammatory properties that are used in Serbian traditional medicine. Many ethnopharmacological studies indicate the use of certain plants for the treatment of diseases related to inflammatory processes, and in this review, we have selected 15 plants for which the anti-inflammatory potential has been confirmed by studies.

## 2. Materials and Methods

After a detailed selection of plants used in the Serbian tradition based on written sources (books) by local authors [11,12,13], the scientific literature that examines the named plants was searched in various scientific databases such as Google Scholar, PubMed, Science Direct, and Web of Science. The keywords include a combination of Latin or English names of selected plants with the words “anti-inflammatory”, “phytochemistry”, and “pharmacology”. The content of selected articles was evaluated to determine their suitability for our topic. Mainly, articles in the English language, published from 2000 to 2023, with full text available were included. The exception is a few articles (books) in Serbian and Russian. These references are published before 2000 [11,12,13,14,15,16]. However, they have been exclusively included because of relevance as they provide a detailed list of all plants used in traditional Serbian medicine and were used to select plants with anti-inflammatory activity. The search was performed from November 2022 to the end of January 2023.

## 3. Anti-Inflammatory Plants from Serbian Traditional Medicine

The use of medicinal plants in Serbia for the treatment of diseases related to inflammatory processes has a long history. The most frequent, in Serbia, inflammation treated with plants, is located at the mucous membrane, the upper respiratory tract, the gastrointestinal tract, and the skin [11,17]. There are several studies dealing with the ethnopharmacology approach of medicinal plant application among Serbian people, mainly in rural areas [18,19,20]. However, plants used as anti-inflammatories in Serbian traditional medicine belong to different families: Asteraceae (*Achillea millefolium*, *Calendula officinalis*, *Matricaria chamomilla*), Rosaceae (*Agrimonia eupatoria*), Poaceae (*Agropyrum repens*), Liliaceae (*Allium cepa*, *A. sativum*), Malvaceae (*Althaea officinilas*), Betulaceae (*Betula pendula*), Fabaceae (*Glycyrrhiza glabra*), Juglandaceae (*Juglans regia*), Hypericaceae (*Hypericum perforatum*), Lamiaceae (*Mentha piperita*, *Salvia officinalis*), and Salicaceae (*Salix alba*) (Figure 1).

### 3.1. Achillea millefolium L., Asteraceae (Eng. Yarrow, Srb. Hajdučka Trava)

Yarrow is one of the most famous and most commonly used plants in traditional Serbian medicine. The internal and external use of this plant in which people have great confidence is widespread. In the form of poultices and ointments, it is used by people to treat various inflammatory and injured skin conditions. This action was confirmed by in vitro and in vivo studies in which extracts of yarrow showed a significant anti-inflammatory effect as well as a healing effect. In addition to independent use, it is also used as part of complex herbal mixtures for various medicinal purposes [13,14,21]. The data obtained in one in vivo study showed that the oil yarrow extracts had a significant anti-inflammatory property. The application of tested oil extracts on the artificially irritated skin of volunteers demonstrated the ability to re-establish their optimal skin parameters to the values measured prior to the irritation. The topical anti-inflammatory activity is attributed to the sesquiterpenes being caused by their inhibition of the arachidonic acid metabolism [22]. Internally, yarrow is part of tea mixtures intended for the treatment of asthma as chronic inflammation of the respiratory tract [14]. One study aimed to evaluate the ethanolic extract of *A. millefolium* flower for antitussive and anti-asthmatic potential through animal experimental models. The results of the study revealed the potent antitussive and antiasthmatic activities of *A. millefolium* flower extract [23].

### 3.2. Agrimonia eupatoria L., Rosaceae (Eng. Common Agrimony, Srb. Petrovac)

Common agrimony, church steeples, or sticklewort is another important herb of traditional medicine with numerous uses. Aerial parts of this plant are used internally in the form of tea or externally in the form of baths. It is known to be used for painful joints and inflammatory diseases of the mouth and throat [13,14]. Anti-inflammatory effects are well studied and confirmed in a lot of in vitro and in vivo studies. Experiments showed that *A. eupatoria* exerts an immunoprotective effect and decreases the levels of pro-inflammatory cytokines while increasing those of anti-inflammatory cytokines [24,25]. Antioxidant, anti-inflammatory, and peripheral analgesic activities were observed for *A. eupatoria* infusion and polyphenol-enriched fraction. Based on the results of this study, it was concluded that the traditional use of the *A. eupatoria* infusion as an antioxidant and anti-inflammatory is justified and suggests that its polyphenols (isoquercetin, tiliroside, and kaempferol O-acetyl-hexosyl-O-rhamnoside) contribute to this activity and should be considered as lead molecules for designing new pharmacophores [26,27].

### 3.3. Agropyrum repens L., Poaceae (Eng. Couch Grass, Srb. Pirevina)

Couch grass is a very common perennial species of grass native to most parts of Europe. For medicinal purposes, the rhizome is used to make tea for the treatment of various inflammatory conditions. Traditional use has been recorded for inflammation of the bladder, bronchitis, arthritis, and rheumatism [13,14]. One study showed that oral administration of ethanol extract of rhizomes of *A. repens* induced moderate inhibition of carrageenan foot oedema of the rat hind-paw compared to indomethacin. In the other study, the cream containing dry couch grass extract was tested for allergic contact dermatitis in rats. The results showed that the anti-inflammatory effects of the couch grass cream were comparable to the standard glucocorticoid cream activity [28].

### 3.4. Allium cepa L., Liliaceae (Eng. Onion, Srb. Crni Luk)

Onion is widely used in Serbia both for therapeutic purposes and as a spice and part of traditional cuisine. The whole plant is edible, but the bulbs that grow underground are most commonly used. It is used in fresh and dried form as a spice in food and in the form of teas or poultices for the treatment of various diseases, including inflammatory conditions such as headache, common cold, arthritis, and asthma. Externally, onion juice or juice is used in the treatment of skin inflammation, purulent wounds, burns, frostbite, and insect bites [13,14]. Numerous modern research has confirmed the justification of the use of onion as an anti-inflammatory agent. It was reported that the anti-inflammatory properties of *Allium* species are due to the presence of effective compounds such as tannin, flavonoids, anthocyanin, saponin, etc. Thiosulfinates and cepaenes from onion showed anti-inflammatory properties mediated through the inhibition of chemotaxis of human polymorphonuclear leukocytes. Furthermore, it has been established that cepaenes inhibit cyclooxygenase (COX) and lipoxygenase (LOX) enzymes. Quercetin, a well-known constituent of onion decreased the production of inflammatory cytokines such as IL-1a, IL-4, and TNF-a and inhibited the proliferation and activity of lymphocytes. These effects have been confirmed through several studies on animals and humans [29,30,31].

### 3.5. Allium sativum L., Liliaceae (Eng. Garlic, Srb. Beli Luk)

Garlic is another valuable plant from the genus *Allium* which is very widespread. Among people, garlic is a medicine for all ailments, in which there is a great and unshakable trust. This belief often goes so far that garlic is still used today in some households in the countryside not only as a preventative but also as a protective agent against “evil spirits” and other dangerous “invisibles”, for fortune-telling, recovery, spells, and magic. The primordial belief in the medicinal, protective, and magical power of garlic left a deep mark on the material and spiritual life of the people. Garlic is eaten almost regularly and is added as an ingredient to various dishes. During epidemics of typhus, cholera, plague, dysentery, flu, and in general, whenever there was a great plague of infectious diseases, garlic was always recommended and used daily as a preventive and curative tool [13,14]. Garlic extracts and their related phytochemicals have been reported to possess anti-inflammatory activity in numerous studies [32]. Allicin, the main constituent of garlic, demonstrated a defensive mechanism against pathogens by its ability to enhance the activity of immune cells and influence signaling pathways associated with these immune cells. Moreover, allicin works on T-cell lymphocytes by inhibiting the SDF1α chemokine which is associated with the weakness of the dynamic structure of the actin cytoskeleton in addition to this, it leads to inhibit the transendothelial migration of neutrophils [33]. Another report indicated that thiacremonone (a sulfur compound isolated from garlic) prevents neuroinflammation and amyloidogenesis by blocking the nuclear factor kappa-light-chain-enhancer of activated B cells (NF-κB) activity, and for that reason can be used to treat neurodegenerative disorders related to inflammation [34].

### 3.6. Althaea officinilas L., Malvaceae (Eng. Marshmallow, Srb. Beli Slez)

Marshmallow is a widespread plant in Serbia. All parts of the plant can be used for treatment, but the root, which is the richest in active principle, is used most often. Macerate of white marshmallow is used as an auxiliary mucus agent for inflammation of the respiratory tract and gastroenteritis, and as compresses for inflammation of the skin. Cultivated plants are generally used to obtain plant raw materials because they are of better quality with a higher content of active principles [13,14]. The in vitro experiments on human monocytic cell line THP-1 showed a significant anti-oxidant and anti-inflammatory activity of root extracts of *A. officinalis.* The investigated preparation showed the ability to ameliorate the migratory capacity of macrophages. These anti-inflammatory effects were comparable to or even better than diclofenac [35].

### 3.7. Betula pendula Roth., Betulaceae (Eng. Common Birch, Srb. Bela Breza)

Common birch is widely distributed in temperate and northern climate zones. It is used in the traditional medicine of numerous countries, and its positive effects on human health have been known since ancient times. It belongs to the group of medicinal plants. Numerous studies on the chemical composition and activities of birch isolates aim to confirm their biological effects and use in traditional medicine. Birch leaf is one of the main ingredients of tea mixtures for the treatment of inflammation of the urinary tract acting as a diuretic and does not irritate the renal parenchyma. Birch tar is also used in folk medicine, a substance derived from the dry distillation of the wood, bark, and roots of the birch tree. It is used in dermatology, especially in eczema as a condition of chronic inflammation of the skin. In addition to tea and tar, birch sap is also used, a colorless liquid that oozes in the spring from cut birch trees [13,14]. One study investigates the influence of the aqueous extract of *Betula pendula* on primary human lymphocytes in comparison to the synthetic anti-arthritis drug methotrexate in vitro on human peripheral blood mononuclear cells (PBMC). These results provide a strong rational base for the widespread use of the leaf extract of *Betula pendula* in the treatment of immune disorders such as rheumatoid arthritis, through the reduction of proliferating inflammatory lymphocytes [36].

### 3.8. Calendula officinalis L. Asteraceae (Eng. Calendula, Srb. Neven)

Calendula is native to the Mediterranean region but it is widely cultivated all over Serbia. It has a very wide application in traditional medicine. It is used in the form of tea, as an addition to salads and other dishes, and externally in the form of oils, ointments, compresses, or rinsing teas. For curative purposes, inflorescences are used and rarely is the above-ground part of the plant in bloom. Traditionally, many beneficial effects are attributed to this plant. It is used as an anti-inflammatory and a remedy for healing wounds and skin disorders. Calendula ointments, suspensions, or tinctures are used topically for treating acne, reducing inflammation, controlling bleeding, and soothing irritated tissue [13,14]. Numerous studies confirm the anti-inflammatory effect of this plant applied topically. One in vitro study assessed the anti-inflammatory potential of calendula oil using lipopolysaccharide (LPS)-stimulated macrophages, as an in vitro model of inflammation. Scientists investigated the ability of a commercial calendula flower extract to inhibit NO production on macrophages exposed to LPS. The obtained results showed a dose-dependent NO inhibition of up to 50%, presenting a safety profile, thus, reinforcing the anti-inflammatory activity of calendula flower extract. In conclusion, the results of this study support the usefulness of Calendula oil in the treatment of injured skin and for conditions or diseases for which NOS contributes to the pathophysiology, such as contact dermatitis, vitiligo, rosacea, melasma, and psoriasis [37]. A study on rats provided evidence that *Calendula officinalis* presented anti-inflammatory and antibacterial activities as well as the capability of stimulating fibroplasia and angiogenesis. Calendula extracts showed a positive effect on the inflammatory and proliferative phases of the healing process of cutaneous wounds in rats [38]. In addition, the topical application of *C. officinalis* ointment has helped to prevent dermatitis and pain, thus, reducing the incidence rate of skipped radiation treatments in randomized trials [39].

### 3.9. Glycyrrhiza glabra L., Fabaceae (Eng. Liquorice, Srb. Sladić)

Liquorice is a famous medicinal plant worldwide. The root is the most used part of this plant with an extremely sweet flavor and pleasant odor. Liquorice has had an important place in traditional Serbian medicine since ancient times. In monastery hospitals, tea made from a mixture of liquorice root, rhizomes, and barley was used as a universal remedy for reconvalescence. Modern research has also confirmed its beneficial effect on the liver, so licorice root is a common ingredient in detox tea mixes. The active ingredients show anti-inflammatory action and have a beneficial effect on spasms and pain relief. Liquorice is part of several tea mixtures with therapeutic usages, namely *Species pectorales, Species diureticae*, and *Species urologicae*) and it is also part of a tea mix for the pediatric population. Black sugar is a product of liquorice made from the aqueous extract by steaming to dryness, which contains around 25% glycyrrhizin [13,14]. Many studies have investigated the anti-inflammatory effect of this plant and the results have confirmed the justification of its use in tradition. Some studies concluded that glycyrrhetinic acid and aqueous extract of liquorice possess strong anti-inflammatory activity, which was comparable with diclofenac [40,41]. Additionally, it was further recommended that the activity of anti-inflammatory formulations such as famotidine or diclofenac can be further enhanced through the addition of liquorice aqueous extract [41]. A lot of studies evaluated the impact of *G. glabra* and its bioactive components on different mechanisms of inflammation. Results showed inhibition of proinflammatory cytokine through inhibition of LPS-induced IL-1β, IL-6, IL-8, and TNF-α responses of macrophages. Furthermore, one study showed that extract of this plant inhibits serum levels of TNF-α and reduces antigen induce arthritis symptoms in mice [42].

### 3.10. Juglans regia L. Juglandaceae (Eng. Walnut, Srb. Orah)

Walnut is an ancient plant that is cultivated and grows wild in Serbia. In addition to the nutritional value of the fruit, the leaf and pericarp of the young fruit are traditionally used in healing. Externally, walnut-leaf tea is used to rinse the skin and mucous membranes in various inflammatory processes. Walnut tea is drunk orally for inflammation of the mucous membrane of the digestive organs. Furthermore, walnut leaves are added to tea mixtures for improving the resistance of the body’s immunity [13,14]. The justification of traditional use has been confirmed by research. One study evaluated the antitussive, antioxidant, and anti-inflammatory effects of a walnut extract rich in bioactive compounds, using a citric acid-induced cough model in rats. Walnut septum showed significant antitussive and anti-inflammatory activities [43]. The ethanolic extracts of *J. regia* leaves exhibited potent anti-inflammatory activity comparable with indomethacin against carrageenan-induced hind paw edema model in mice without inducing any gastric side effects [44].

### 3.11. Hypericum perforatum L. Hypericaceae (Eng. St. John’s Wort, Srb. Kantarion)

St. John’s wort is a very common plant in the flora of Serbia. The aerial part of the plant is used for a variety of external and internal uses. Traditional written sources mention the use of St. John’s wort in the form of an infusion or tincture as an anti-inflammatory, styptic, and antiseptic agent. St. John’s wort oil, which is made as a macerate with sunflower or olive oil, is a well-known and valued remedy for healing wounds and burns [45]. An important note with the traditional use of the preparation is that the chemical compounds in the composition cause photosensitization, so precaution must be taken when exposed to UV light during treatment [15]. Imaninum is traditional antibacterial preparation for external application based on *H. perforatum* used for the treatment of fresh wounds, burns, and ulcers [13,14]. Imaninum is described as a dark brown powder obtained by boiling ground aerial parts of plant material (without stem) in 10% NaOH (1:10). The NaOH is removed and the cooking is repeated 5–6 times, always with a new amount of NaOH. The herbal residue that remains after cooking is acidified with HCl to an acidic reaction and then grind to a fine powder. In addition, if necessary, chlorophyll and pigments are separated from this preparation. It is used as an external agent in the form of solutions, ointments, and powders for the treatment of patients with fresh and infected wounds, burns, ulcers, abscesses, mastitis, carbuncles, boils, etc. They are also used for acute rhinitis, pharyngitis, laryngitis, and sinusitis [16].

Many in vitro and in vivo studies have justified the traditional use of St. John’s wort in the therapy of inflammatory skin disorders by proving that the lipophilic extract as well as the pure individual components of its composition possess notable anti-inflammatory potential [46]. The study on rats showed that *H. perforatum* decreased levels of enzymes associated with colonic inflammation [47]. Results from a study on rats demonstrated that *H. perforatum* exhibits antiedematogenic and antinociceptive properties, which may be of value for the management of inflammatory painful conditions. However, the side effect is gastric irritation [48].

### 3.12. Matricaria chamomilla L. Asteraceae (Eng. Chamomile, Srb. Kamilica)

Medicinal properties of chamomile have been known since ancient times: in folk medicine, chamomile is considered a “panacea”—a cure for all diseases, given that its primary medicinal properties are antiseptic and anti-inflammatory, whether it is infections or inflammatory processes on the skin, mucous membranes of the mouth and throat or mucous membranes of the respiratory organs, digestive organs, or urogenital system [13,14]. Results from many studies confirmed traditional uses. The result from one study on animals showed that the volatile essential oil and non-volatile components (the aqueous extract and flower-water of chamomile) could significantly inhibit pedal swelling induced by carrageenan in rats, and ear swelling induced by xylol in mice [49]. Moreover, extracts provoked the increase of celiac capillary vessel permeability induced by HAC (glacial acetic acid dissolved in normal saline) in mice and the concentration increase of PGE2 and NO during pedal swelling induced by carrageenan in rats, as well as heterogeneity passive skin allergy in mice’s ear and the inching reaction caused by dextran in mice [50,51]. As expected, the essential oil had the most remarkable anti-inflammatory and antiallergic effects [49]. An animal study showed that bisabolol from chamomile reduces inflammation and fever and has a favourable effect as adjuvant therapy in arthritis [49,52].

### 3.13. Mentha piperita L. Lamiaceae (Eng. Peppermint, Srb. Nana)

The peppermint leaf has multiple uses in Serbian tradition. Mint is a common ingredient in various tea blends, including tea blends for children which confirms the people’s trust in this plant. In addition to consumption for medicinal purposes, mint tea is also drunk as a beverage due to its pleasant taste. Peppermint has gained the trust of folk medicine as an effective and safe tool with no restrictions for any type of use in all age populations. Later, based on the evidence, this precious plant was given generally recognized as safe (GRAS) status by the Food and Drug Administration (FDA). The essential oil of this plant is widely used for various purposes. Externally, it is used as part of the preparation for rubbing against rheumatic and neuralgic pains. By the way, the largest quantities of essential oil are used in the food, cosmetic, and alcoholic beverage industries. It is also used for the extraction of menthol [13,14]. The essential oil and extract of *M. piperita* were evaluated for their in vitro and in vivo anti-inflammatory activity. The investigations showed that the oil of *M. piperita* exerted significant anti-inflammatory activity, without inducing any apparent acute toxicity or gastric damage as compared to indomethacin, as the reference drug [53,54].

### 3.14. Salix alba L. Salicaceae (Eng. White Willow, Srb. Bela Vrba)

White willow has been widely used in traditional Serbian medicine since ancient times. In the treatment, the young bark is used, which is peeled in early spring. Willow bark decoction is used for colds, flu, and rheumatic diseases due to its analgesic, antipyretic, and anti-inflammatory effects. In the form of oral rinses, a decoction of willow bark is recommended for the treatment of inflammatory conditions of the mucous membranes. Until the synthetic production of salicylic acid was perfected, this plant and related willow species were long used as a raw material from which this acid was obtained [13,14]. Contemporary research has confirmed the experiences of folk medicine. In addition to the content of salicylic acid also, the presence of flavonoids is responsible for the anti-inflammatory effects of willow bark extracts, so to achieve a healing effect in the case of lower back pain relief, much lower doses are needed than for aspirin-based treatment [55]. This fact is very important from the aspect of safety of use. Experimental animal models showed that *S. alba* possesses an anti-inflammatory effect in xylene-induced ear oedema or carrageenan-induced paw oedema [56,57].

### 3.15. Salvia officinalis L. Lamiaceae (Eng. Sage, Srb. Žalfija)

Sage is an ethereal, luxurious, medicinal, and Mediterranean plant that has been cultivated for centuries for its healing properties. The most serious diseases were treated with sage before the era of antibiotics, so it could be said that it was the only salvation in those cases. It was prepared and used in different ways. It was used fresh and dry, it was pressed and essential oil was made from it, sage tea was brewed, it was chewed fresh for toothaches and diseased gums, and bandaged on wounds and injuries received in battles. Its exceptional medicinal properties, given that it is one of the strongest natural antibiotics, antimycotics, and antiseptics, rank sage even today in the first place, as the queen among medicinal herbs. In Serbia, people have great confidence in this plant as one of the most important medicinal plants [13,14]. Pharmacological studies have shown that *S. officinalis* has anti-inflammatory and antinociceptive effects. For example, it has been shown that this plant helps to control neuropathic pain in chemotherapy-induced peripheral neuropathy [58]. Among different extracts of *S. officinalis*, the chloroform one shows more anti-inflammatory action, while the methanolic extract and essential oil demonstrate low action [59]. Flavonoids and terpenes are the compounds that most likely contribute to the anti-inflammatory and antinociceptive actions of the herb. One study reported that flavonoids extracted from *S. officinalis* reduce inflammation in the mouse carrageenan model and induce an analgesic effect in a dose-dependent manner [60]. Investigation of individual constituents of *S. officinalis* showed that topical application of rosmarinic acid inhibits epidermal inflammation [61]. Manool, carnosol, and ursolic acid are terpenes/terpenoids with anti-inflammatory potential [62]. The anti-inflammatory action of ursolic acid is significantly more potent than that of indomethacin [59]. This proven action of *S. officinalis* constituents may be responsible for its high value as an anti-inflammatory agent [63].

## 4. Scientific Data on Anti-Inflammatory Potential of Selected Plants

All the above plants have a long history of traditional use as medicinal agents by many peoples around the world. That application is founded and confirmed by experience about the beneficial effect of their application. Modern research tends to investigate and document the justification of their use, explain the mechanisms of action, and isolate the active principles. Various plant-derived compounds inhibit inflammation through a reduction in the levels of several cytokines including IL-1β, IL-6, and TNF-α, and the suppression of COX-2, prostaglandins, and nitric oxide (NO) release. Active organosulphur compounds in garlic primarily ajoene, alliin, and allicin work by reducing levels of pro-inflammatory cytokines while increasing levels of anti-inflammatory interleukins. Different subclasses of flavonoids have been shown to suppress inflammatory molecules such as TNF-α, IL-1, IL-6, IL-17, and IFN-γ, which are secreted through the activation of several signaling pathways, predominantly the NF-κB pathway [9]. The ultimate goal of all research is to provide knowledge for the safe and effective use of the whole plant as an herbal remedy, as well as to find new raw materials for the isolation of bioactive substances for direct use as drugs or starting substances for further chemical modification to improve activity and/or reduce toxicity. Nonetheless, herbal preparations can be used as adjuvant therapy in addition to conventional therapy. In favor of plants and all the advantages of their use, it is worth noting that molecular diversity is a valuable advantage in relation to synthetic molecules because it allows acting on more different molecular mechanisms with lower doses and fewer adverse effects.

An overview of the scientific literature on plants with traditional use in Serbia in the treatment of inflammation is given in Table 1.

Medicinal plants could be a powerful source of raw material for the pharmaceutical industry and synthesis of new remedies, considering that every nation has its own heritage in ethnobotany and ethnopharmacology, so traditional medicine is key important in the processes of plant-based medicines products [83,84]. Taking into account the diversity of the plants, which is conditioned by numerous factors such as geographical, pedological and climatic, intensive research on the bioactive potential of medicinal plants in each region should be the focus of scientists around the world [85,86]. The medicinal plants with anti-inflammatory properties used among Serbian people could be promising candidates for further research and identifying new bioactive potentials, especially in combination with modern green technologies [87,88,89].

## 5. Conclusions

People in Serbia have relied on healing with various herbs since ancient times. Most of the plants from the natural treasures of Serbia found traditional use. In the therapy of inflammation, plants from Asteraceae and Lamiaceae families are most frequently used. The easy availability of numerous herbs and empirically proven effectiveness with high safety are the reason for their popularity in tradition. Today, through in vitro and in vivo tests, the justification for the use of traditional plants is confirmed. With modern knowledge, the best possibilities for using plant potential and their safe application are reached. Where the limitations of conventional medicine are due to side effects, herbal treatment could gain its full recognition due to fewer side effects, complex composition, and synergistic action of individual components. In summary, all those facts open the possibility of using the natural plants’ treasure for the purpose of isolating the active principles and further clinical trials which would relive the reveal benefits and limitations of their application as well as the potential synergistic effect with conventional therapy.

## Figures and Tables

**Figure 1 life-13-00874-f001:**
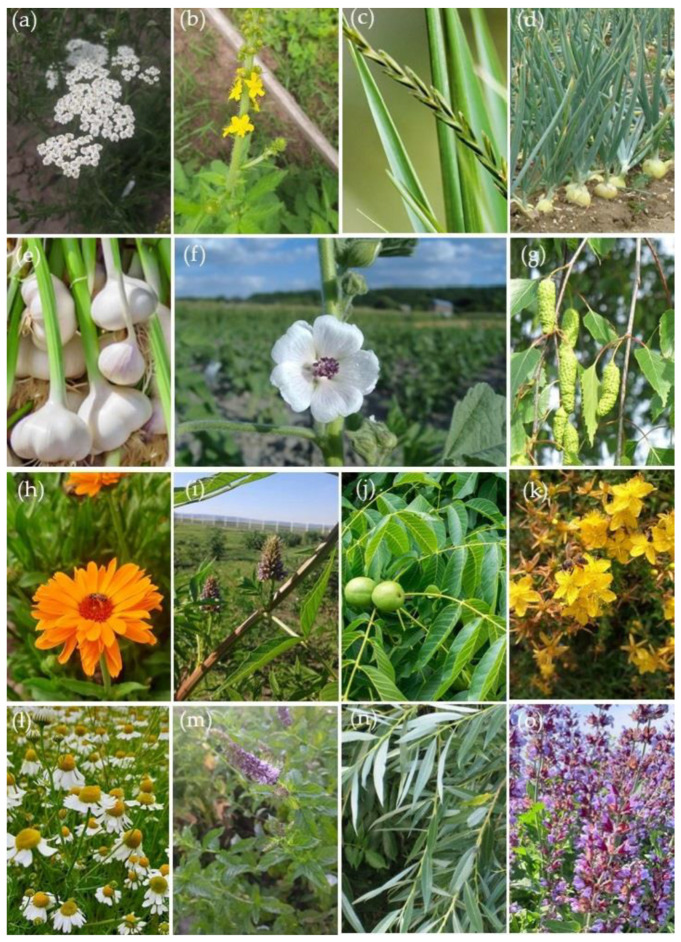
Plants used as anti-inflammatory in Serbian traditional medicine: (**a**) Achillea millefolium; (**b**) Agrimonia eupatoria; (**c**) Agropyrum repens; (**d**) Allium cepa; (**e**) Allium sativum; (**f**) Althaea officinilas; (**g**) Betula pendula; (**h**) Calendula officinalis; (**i**) Glycyrrhiza glabra; (**j**) Juglans regia; (**k**) Hypericum perforatum; (**l**) Matricaria chamomilla; (**m**) Mentha piperita; (**n**) Salix alba; (**o**) Salvia officinalis.

**Table 1 life-13-00874-t001:** Mini-review of scientific data about the anti-inflammatory effect of plants from Serbian tradition.

Name of Herb	Part Used	Phytoconstituents	Scientific Data	Ref.
*Achillea millefolium*	aerial parts	alkaloids, glucoside, choline, essential oils, salicylic acid, sesquiterpenoids, dicaffeoylquinic acids, luteolin, apigenin	down-regulating the expression iNOS inhibition of the inflammation-related proteases, namely, HNE and MMP-2 and MMP-9 macrophage activation modulating agents in vivo study on humans showed the ability to re-establish optimal pH and hydration of the skin to the values measured prior to the irritation significant antitussive and antiasthmatic activity on mice	[21,22,23,64]
*Agrimonia eupatoria*	aerial parts	carbohydrates, tannins, terpenoids, flavonoids, agrimony lactone, glycosides and oils; polysaccharides, triterpenoids, silicic acid, salicylic acid, nicotinamide complex, thiamine, and vitamin K.	inhibiting inflammatory cytokine (IL-1β, IL-4, IL-6) and INF-β production inhibition of NO and PGE2 production significantly reduced carrageenan-induced paw oedema decreases the levels of pro-inflammatory cytokines while increasing those of anti-inflammatory cytokines. improving human markers of lipid metabolism, oxidative status, and inflammation after one month’s consumption in healthy volunteers	[24,25,26,27,28]
*Agropyrum repens*	rhizome	carbohydrates, mucilages, saponins, essential oils	moderate inhibition of carrageenan-induced foot oedema of the rat anti-inflammatory activity impacting plasma lipid peroxidation parameters MDA, DC, and catalase activity	[28]
*Allium cepa*	bulb	Phenolic acids, thiosulphates, and flavonoids	modulating COX-mediated prostaglandin production reducing lung inflammatory cytokines such as IL-4, 5, and 13 and T helper 2 significantly reduced total WBC and lung inflammatory cells such as neutrophil, eosinophil, and monocyte counts, but led to a significant increase in lymphocyte counts in asthmatic Wistar rats	[30,31,65]
*Allium sativum*	bulb	Sulphur compounds, enzymes, amino acids, minerals	modulating leucocyte cell proliferation and cytokine production inhibiting Th1 and inflammatory cytokines and upregulating IL-10 production	[32,66,67,68]
*Althaea officinilas*	root, leaf, flower	starch, pectins, saccharose, mucilage, flavonoids, caffeic acid, p-coumaric acid, isoquercitrin, coumarins, phytosterols, tannins, amino acids	in vitro Phytohustil^®^ and root extracts of A. officinalis were able to protect human MΦ against H_2_O_2_-induced cytotoxicity and H_2_O_2_-induced ROS production. inhibition of the LPS-induced release of TNF-α as well as of IL6 in MΦ. results of a pilot active-controlled trial revealed that topical use of 1% ointment contained ethanolic extract in petrolatum base which has a higher efficacy in children with atopic dermatitis in comparison to topical hydrocortisone 1%.	[35,69]
*Betula pendula*	leaves and leaf buds	flavonoids, tannins, resins, essential oils	in vitro reduction of proliferating inflammatory lymphocytes in vivo study on rats showed a significant effect of tablets with dense extract on carrageenin-induced inflammation	[36,70]
*Calendula officinalis*	inflorescence	triterpenoids, flavonoids, coumarins, quinones, essential oils, carotenoids, and amino acids.	dose-dependent NO inhibition reinforcing the anti-inflammatory activity of calendula flower extract. potent anti-inflammatory response extract may be mediated by the inhibition of proinflammatory cytokines and Cox-2 and subsequent prostaglandin synthesis. methanolic extract of flowers showed the most potent inhibition of TPA-a-induced inflammation in mice antioedematous effect in croton oil-induced mouse oedema cream containing calendula extract has been reported to be effective in dextran and burn oedemas as well as in acute lymphoedema in rats. reducing inflammatory bone resorption in an experimental rat periodontitis in vivo on rats aqueous flower extract demonstrated anti-inflammatory and antibacterial activities as well as the capability of stimulating fibroplasia and angiogenesis phase III Randomized Tria showed that calendula-based ointment was statistically significantly more effective than trolamine in preventing acute dermatitis grade 2 or higher during adjuvant postoperative breast irradiation.	[38,39,71,72,73,74]
*Glycyrrhiza glabra*	root	triperpenic saponins, sterols, flavonoids	the hydroalcoholic extract showed a maximum inhibitory action on carrageenan-induced paw oedema at the dose of 200 mg/kg and inhibited the leukocyte migration in a dose-dependent manner. The anti-inflammatory activity was comparable to indomethacin methanol extract was evaluated for the COX-2 inhibitory activity using Cayman COX (ovine) inhibitory screening assay. A few molecules (dominantly glycyrrhizic acid) showed potent COX-2 inhibitory activity which may be beneficial as anti-inflammatory agents glycyrrhizin exhibited steroid-like anti-inflammatory activity, comparable with hydrocortisone, due to inhibition of phospholipase A2 activity, glycyrrhizic acid inhibited cyclooxygenase activity and prostaglandin formation (specifically PGE2), as well as indirectly inhibiting platelet aggregation outcomes of one in vitro study show that licoflavanone can decrease iNOS and COX2 expression levels in LPS-stimulated RAW 264.7 cells, interfering with the inflammatory cascade mediated by NO and PGE2	[42,75,76,77]
*Juglans regia*	leaves	tannins, naphthoquinone derivatives, flavonoids	in vivo on mice aqueous and ethanolic extracts showed activity against acute and chronic inflammation potent anti-inflammatory activity comparable with indomethacin against carrageenan-induced hind paw edema model in mice without inducing any gastric side effects citric acid-induced cough model in rats showed that walnut septum expressed potent antitussive and anti-inflammatory activities.	[43,44,78]
*Hypericum perforatum*	aerial part	hyperforin, hypericin, flavonoids, tannins	inhibition of the production of PGE2on mouse macrophage cells the compounds isolated from H. perforatum induced a dose-dependent reduction in oedema in mice the results from a study on rats provide evidence for the usage of Oleum Hyperici as an anti-inflammatory and gastroprotective agent	[79,80,81]
*Matricaria chamomilla*	flowers	essential oils, sesquiterpene lactones, coumarins, mucilages	reduction of NO production and induction of anti-inflammatory cytokine production (IL-10) significantly inhibiting swelling of mouse ears caused by xylene, pedal swelling caused by carrageenan in rats, and the increase of celiac capillary vessel permeability in mice inhibitory effect on the increase in PGE2 and NO levels in rat pedal edema caused by carrageenan study on rats showed that chamomile extract prevented a significant increase in serum levels of TNF-α, CRP, IL-6, and fibrinogen. synergic anti-inflammatory effects with diclofenac and indomethacin on carrageenan-induced paw inflammation and stomach damage in rats	[49,51,52]
*Mentha piperita*	leaves	essential oils, phenolics, flavonoids, tannins	potent anti-inflammatory activity in the croton oil-induced mouse ear oedema model inhibitory effect on the production of NO and PGE2	[53,82]
*Salix alba*	bark	phenolic glycosides, flavonoids, tannins, aromatic aldehydes, and acids	in vivo tests of the methanolic and aqueous extracts of the barks showed a strong effect on carrageenan-induced paw oedema and xylene-induced ear edema in vivo on mice methanolic extracts exhibited a dose-dependent analgesic property with more potency than the standard drug aspirin in all tested doses, as well as inhibited the paw edema by interruption of the arachidonic acid metabolism and shows inhibition of the inflammation greater than the inhibitory effect of the aspirin	[55,56]
*Salvia officinalis*	leaves	essential oil, tannins, diterpenes, triterpenes, flavonoids	the chloroform extracts showed strong anti-inflammatory properties on croton oil-induced ear oedema in mice after topical application. in mice, arnosol and ursolic acid/oleanolic acid inhibited the inflammatory phase of formalin and the nociception and mechanical allodynia induced by cinnamaldehyde	[59,62,63]

COX2—cyclooxygenase-2; CRP—C-Reactive Protein; DC—diene conjugates; HNE—neutrophil elastase; IL-6—Interleukin 6; NO—nitric oxide; INF-β—interferon-β; iNOS—inducible nitric oxide synthase; LPS—lipopolyccharide; MDA—malondialdehyde; MMP—matrix metalloproteinases; MΦ—*Macrophages;* PGE2—prostaglandine E2; RAW 264.7—macrophage cell line; ROS—reactive oxygen species; TNF-α—Tumor Necrosis Factor-α; TPA—12-*O*-tetradecanoylphorbol-13-acetate; WBC—white blood cells.

## Data Availability

Not applicable.

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
