# Peer review of "Anti-Inflammatory Properties of Plants from Serbian Traditional Medicine"

_life, 2023, doi:10.3390/life13040874_

Round 1

Reviewer 1 Report

The paper is very interesting, well structured and with a lot of data. What is missing is indicated in the text.

Author Response

The authors of the paper wish to thank our Reviewer for the time and effort invested in evaluating our paper. Comments and guidelines are of great value in our aim to provide an interesting, quality paper on the topic covered in the manuscript. Our group of authors evaluated your comments; given suggestions were accepted and the manuscript was revised.

Reviewer 2 Report

This article presented anti-inflammatory properties of plants from Serbian traditional medicine. It will help the scientific community to understand action of plant molecules against severe diseases. Before recommending this article for publication, the article need substantial major revisions.

Abstract

Generally abstract is short summary of research work where we highlight introduction of our topic, short methodology, results, and conclusion but here the methodology section is very short, void and not clear. The author should explain how they collected data or surveyed literature.

The author should discuss conclusion and future aspects of this research in 2-3 sentences at the end of abstract.

Introduction

The introduction section is well written and provided with good justification background but still there are some deficiencies.

Line 51 should be cited with recent study as well. The following study would be helpful.

https://doi.org/10.3390/antiox12020268

Line 52-58 lack references. Line 55 could be cited with recent study. https://doi.org/10.1016/j.chnaes.2021.03.009

In introduction section specifically in first paragraph causes of inflammation in Serbia should be discuss.

In this review important plants are discussed which are also famous as anti-inflammatory.

A section discussing and presenting mechanism of action of phytochemicals and plant molecules would be effective.

Conclusion is very short and specific some general aspects must be provided.

Also add future perspective and research gap in conclusion.

Author Response

(The authors gave the same response as above.)

Reviewer 3 Report

This review paper is very interesting because it is a good contribution for the knowledge about plants with traditional anti-inflammatory use in Serbia with reference to available studies that support this property. The biological activity of selected plants is attributed to flavonoids, phenolic acids, sterols, terpenoids, sesquiterpenes and tannins.

 The article has one Figure, with fifteen photos of the plants present in the nature and used as anti-inflammatory agents in Serbian traditional medicine, and one Table with a good overview of the anti-inflammatory potential of these plants supported by bibliographic references.

 There is a single minor correction that should be made:

 - On page 13 (line 364), “from the” is repeated;

 With this minor change, the recommendation will be to accept the manuscript for publication.

Author Response

The authors of the manuscript wish to thank our Reviewer for the time and effort invested in evaluating our paper. Our group of authors evaluated your comments; given suggestions were accepted and the manuscript was revised.
